# Modular transcriptional programs separately define axon and dendrite connectivity

Yerbol Z Kurmangaliyev[1†], Juyoun Yoo[2†], Samuel A LoCascio[1†], S Lawrence Zipursky[1]*

[1]Department of Biological Chemistry, Howard Hughes Medical Institute, David Geffen School of Medicine, University of California, Los Angeles, Los Angeles, United States; [2]Neuroscience Interdepartmental Program, University of California, Los Angeles, Los Angeles, United States

**Abstract** Patterns of synaptic connectivity are remarkably precise and complex. Single-cell RNA sequencing has revealed a vast transcriptional diversity of neurons. Nevertheless, a clear logic underlying the transcriptional control of neuronal connectivity has yet to emerge. Here, we focused on *Drosophila* T4/T5 neurons, a class of closely related neuronal subtypes with different wiring patterns. Eight subtypes of T4/T5 neurons are defined by combinations of two patterns of dendritic inputs and four patterns of axonal outputs. Single-cell profiling during development revealed distinct transcriptional programs defining each dendrite and axon wiring pattern. These programs were defined by the expression of a few transcription factors and different combinations of cell surface proteins. Gain and loss of function studies provide evidence for independent control of different wiring features. We propose that modular transcriptional programs for distinct wiring features are assembled in different combinations to generate diverse patterns of neuronal connectivity.

DOI: https://doi.org/10.7554/eLife.50822.001

*For correspondence:
LZipursky@mednet.ucla.edu

†These authors contributed equally to this work

Competing interests: The authors declare that no competing interests exist.

## Introduction

Brain function relies on precise patterns of synaptic connections between neurons. At the cellular level, this entails each neuron adopting a specific wiring pattern, the combination of specific synaptic inputs and outputs. In invertebrates, stereotypical wiring patterns are genetically encoded in the programs regulating the development of neurons. Much of the specificity of inputs and outputs of neurons in the mammalian CNS is also genetically determined (*Sanes and Zipursky, 2010*).

Vast numbers of neurites from a diversity of neurons are intermingled within the developing central nervous system, and they form highly specific synaptic connections with a discrete subset of the neurons they contact. Studies in both vertebrates and invertebrates have led to the identification of cell surface proteins (CSPs) that mediate selective association between neurites (*Tessier-Lavigne and Goodman, 1996*; *de Wit and Ghosh, 2016*; *Zinn and Özkan, 2017*). Gain and loss of function genetic studies have shown that combinations of different CSP families regulate this specificity (*Zarin et al., 2014*). Indeed, neuronal subtypes express highly diverse repertoires of CSPs during circuit assembly (*Tan et al., 2015*; *Li et al., 2017*; *Sarin et al., 2018*). Conserved regulatory strategies involving combinations of transcription factors (TFs) establish unique neuronal identities (*Allan and Thor, 2015*; *Enriquez et al., 2015*; *Hobert, 2016*). However, the programs regulating expression of CSPs for specific neuronal wiring features are still poorly understood.

Single-cell RNA sequencing (RNA-Seq) provides an unsupervised approach to uncover the genetic programs underlying specific wiring features by exploring subtype-specific transcriptomes

during development. As neuronal subtypes exhibit differences in characteristics other than wiring patterns, the relationship between genes and wiring specificity may be obscured by genes contributing to other aspects of neuronal diversity. Therefore, sets of closely related neurons with highly specific differences in wiring patterns are ideally suited to uncover the genetic programs specific to wiring. Here, we explore the genetic logic underlying synaptic specificity in one such set of neurons: T4/T5 neurons of the *Drosophila* visual motion detection pathway. We envision that our findings in this system will provide insights into the genetic logic of wiring specificity more broadly in both vertebrate and invertebrate systems.

T4/T5 neurons share a common developmental origin, physiological function, and general morphology, but differ in their precise wiring patterns and preferred stimulus (*Fischbach and Dittrich, 1989*; *Maisak et al., 2013*; *Apitz and Salecker, 2018*; *Pinto-Teixeira et al., 2018*; *Shinomiya et al., 2019*). There are eight morphological subtypes of T4/T5 neurons in each column of the lobula plate (LoP) neuropil (see below), comprising the most abundant cell type in the fly visual system. These subtypes can be classified into two quartets of subtypes based on dendritic inputs: the four T4 subtypes share a common set of dendritic inputs in the medulla, and the four T5 subtypes share a different set of dendritic inputs in the lobula (*Figure 1A–C*). T4 neurons respond to ON stimuli (i.e. bright edges moving against a dark background) and T5 to OFF stimuli (i.e. dark edges moving across a

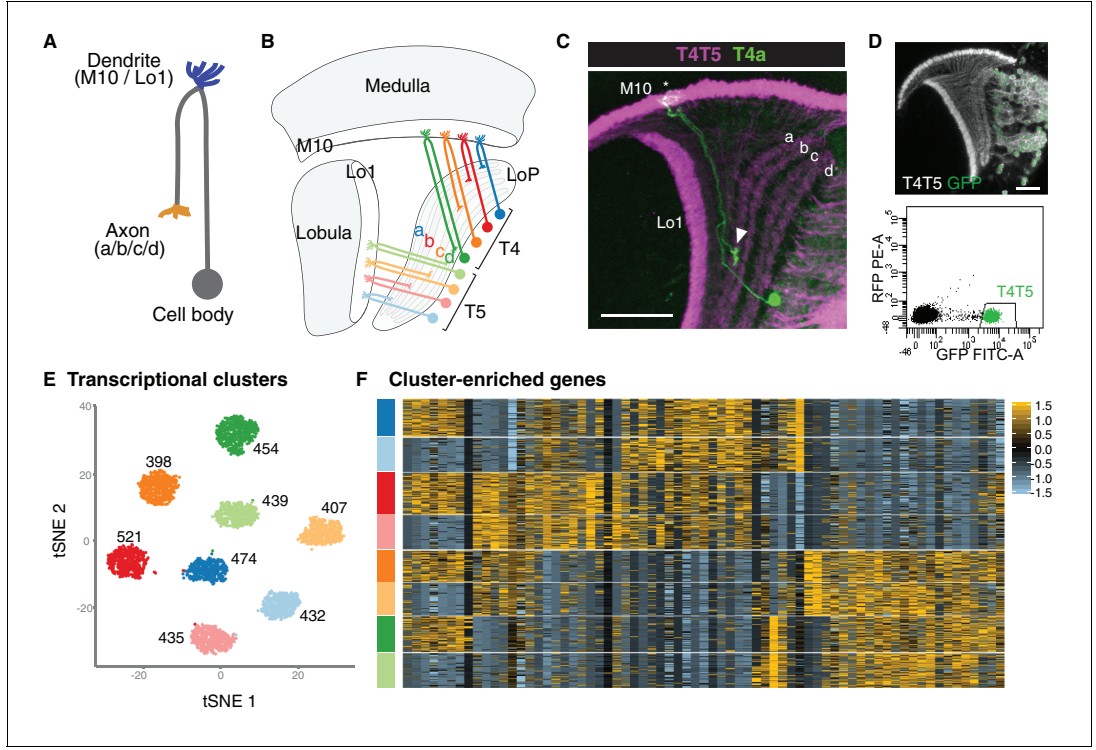

**Figure 1.** Single-cell sequencing reveals eight transcriptionally distinct populations of T4/T5 neurons. (A) Common morphology of a T4/T5 neuron, with axon and dendrite wiring pattern variations in parentheses. (B) Arrangement of the eight T4/T5 subtypes in the optic lobe. Each subtype is defined by a combination of one dendrite (M10 or Lo1) and one axon (LoP a, b, c, or d) wiring pattern. (C) A single T4a neuron (green) with dendrites in M10 (asterisk) and axon terminal in LoP layer a (arrowhead). All T4/T5 neurons labeled in magenta. Scale bar, 20 μm. (D–F) Single-cell sequencing of T4/T5 neurons at 48 hr APF. Unsupervised analysis revealed eight distinct transcriptional clusters. (D) T4/T5 neurons were labeled with nuclear GFP, purified by FACS and used for single-cell RNA-Seq. (E) t-distributed stochastic neighbor embedding (tSNE) plot of 3557 single-cell transcriptomes. Clusters are color-coded according to subtype identity based on following results. Cell numbers are displayed for each cluster. See also *Figure 1—figure supplement 1*. (F) Heatmap of expression patterns of cluster-enriched genes ('one versus all', see Materials and methods). Cells (rows) grouped by cluster identities as in (E). Genes (columns) are ordered by similarity of their expression patterns. Scaled expression levels are indicated, as in scale.

DOI: https://doi.org/10.7554/eLife.50822.002

The following figure supplement is available for figure 1:

**Figure supplement 1.** T4/T5 neurons robustly cluster into eight transcriptionally distinct populations (48 hr APF).

DOI: https://doi.org/10.7554/eLife.50822.003

bright background). T4/T5 neurons can also be classified into four pairs of subtypes (a-d) based on the location of their axon terminals within a given column in layers a-d of the LoP. Each pair responds selectively to visual motion in one of four cardinal directions: posterior, anterior, upwards, and downwards, respectively (*Figure 1A–C*). Although transcriptional profiling of the adult *Drosophila* brain revealed a common transcriptional signature for all T4/T5 neurons, genetic programs for individual subtypes have not been identified (*Davie et al., 2018*; *Konstantinides et al., 2018*). We hypothesized that identification of gene expression programs for individual T4/T5 subtypes during circuit assembly would provide insight into the genetic programs regulating discrete wiring features.

Here, we report that independent transcriptional programs define the dendritic inputs and axonal outputs of T4/T5 neurons. We present gain and loss of function studies indicating that these programs control their corresponding morphological features. Our findings suggest that the modular assembly of separate dendritic and axonal transcriptional programs contributes to the diversity of wiring patterns in complex nervous systems.

## Results

### Single-cell RNA-Seq reveals eight transcriptionally distinct populations of T4/T5 cells

As a step towards uncovering genetic programs that control neuronal wiring patterns, we performed single-cell RNA-Seq on developing T4/T5 neurons. Sequencing was performed at 48 hr after puparium formation (APF). This developmental time point precedes a period of widespread synaptogenesis in the visual system, and coincides with the appearance of four discrete synaptic layers (a, b, c, d) in the LoP neuropil. Neurons were purified from dissected optic lobes by FACS using a transgenic line with nuclear GFP selectively expressed in all T4/T5 neurons (*Figure 1D*). RNA-Seq libraries were generated using 10X Chromium technology (*Zheng et al., 2017*) and sequenced to a mean depth of 92,000 reads per cell. In total, we profiled 3894 cells with a median of 1633 genes and 4389 transcripts captured per cell. After quality control and removal of outlier cells, our final dataset consisted of 3557 cells with 1000–2000 genes per cell.

We applied independent component analysis (ICA) followed by a graph-based clustering method to separate transcriptionally distinct cell populations (*Butler et al., 2018*; *Saunders et al., 2018*). Unsupervised analysis revealed eight clusters of approximately equal numbers of cells (*Figure 1E* and *Figure 1—figure supplement 1*), suggesting that each cluster corresponded to a single T4/T5 subtype.

To identify genes preferentially expressed in T4/T5 subtypes, we performed differential gene expression analysis between each of the eight individual clusters and all other cells in the dataset (i.e. 'one versus all,' see Materials and methods). This revealed 69 genes which were strongly expressed in some clusters and not in others. Cluster-enriched genes, however, were not specific to single clusters. By contrast, for instance, each of the five subtypes of lamina neurons is defined by at least one subtype-type specific transcription factor (*Tan et al., 2015*). Thus, while T4/T5 subtypes separated into eight transcriptionally distinct clusters, they were not defined by unique molecular markers (*Figure 1F*).

### Eight T4/T5 transcriptional clusters are separated by three primary axes of transcriptional diversity

The absence of unique markers for individual subtypes suggested that they were instead defined by unique combinations of genes. ICA has been shown to capture groups of genes corresponding to discrete biological phenomena (*Saunders et al., 2018*). Intriguingly, three independent components (ICs) each split the eight T4/T5 clusters into two groups of four, each in a different way (*Figure 2A*). Together, these three ICs were sufficient to define all eight clusters (*Figure 2B*).

Each of the three ICs defined an axis of transcriptional diversity (hereafter referred to as Axis 1, 2, 3) driven by a group of genes differentially expressed along each axis. Many of these genes were expressed in a binary (ON/OFF) pattern in one of the two groups of clusters separated along each axis (*Figure 2C*). A small set of TFs were among the genes with the highest contributions to each axis and illustrate this pattern. Binary expression of *bifid* (*bi*) defined the two groups of clusters separated by Axis 1, *grain* (*grn*) defined the clusters separated by Axis 2, and *TfAP-2* defined the clusters

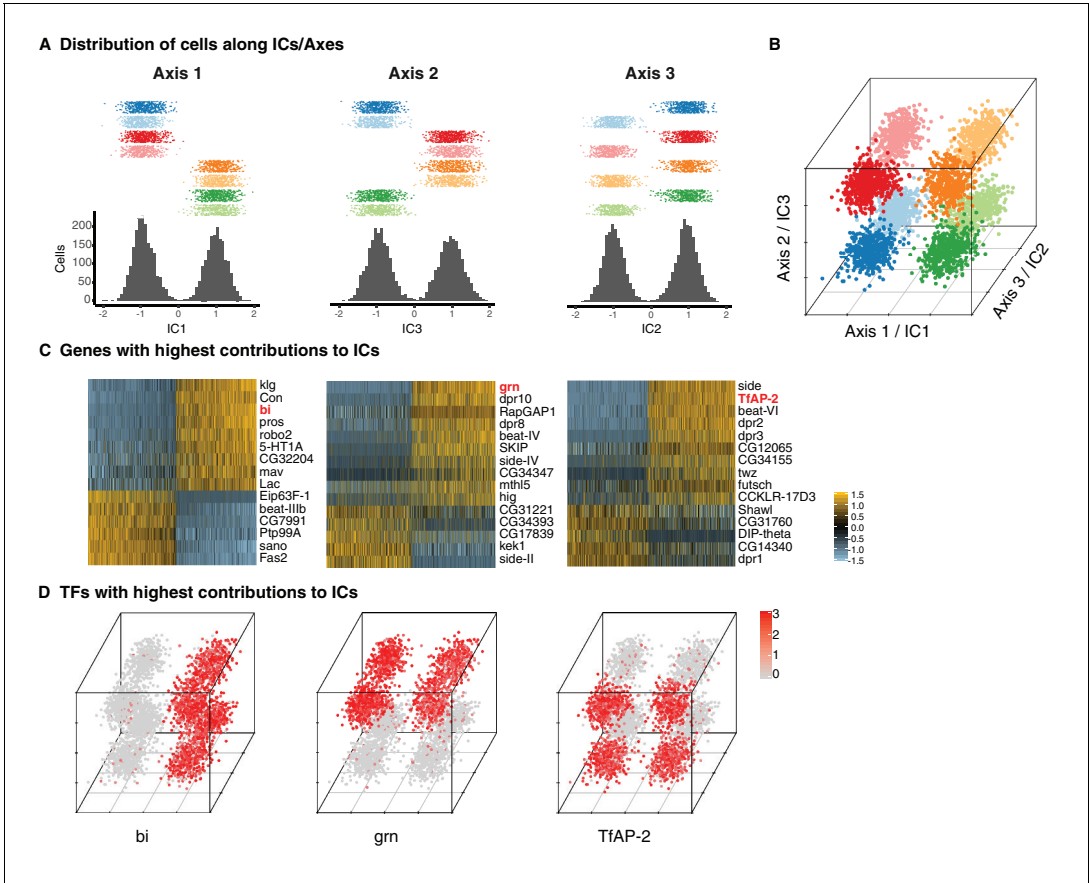

**Figure 2.** Three primary axes of transcriptional diversity define eight T4/T5 populations. (**A**) Three independent components (ICs, henceforth Axis 1, 2, 3) separate cells into approximate halves. Histograms (bottom) and 1-D scatterplots (top) show the distributions of cells along each axis. Cells are grouped into rows based on cluster identities. ICs/Axes are ordered according to following results. Clusters are color-coded as in *Figure 1E*. See also *Figure 1—figure supplement 1*. (**B**) 3-D scatterplot of the distributions of cells along the three ICs/Axes. (**C**) Heatmaps of expression patterns of the top 15 genes with highest contribution (loading) to each IC/Axis. Cells (columns) are ordered according to a score for each IC/Axis. Genes (rows) are ordered according to the contribution to each IC/Axis. Scaled expression levels are indicated, as in scale. Axes ordered as in (**A**). (**D**) 3-D scatterplots with expression patterns of transcription factors (TFs) with highest contribution to each IC/Axis. Normalized expression levels are indicated by color, as in scale. Axes are arranged as in (**B**).

DOI: https://doi.org/10.7554/eLife.50822.004

separated by Axis 3 (*Figure 2D*). Thus, while no individual cluster is uniquely defined by the expression of a single gene, each cluster expresses a unique combination of genes. In this way, three axes of diversity with orthogonal ON/OFF expression patterns of TFs define the eight T4/T5 clusters.

## Primary axes of transcriptional diversity correspond to axon and dendrite wiring patterns

We next sought to map transcriptional clusters to T4/T5 subtypes, and to determine the biological significance of the observed axes of transcriptional diversity. We inspected in vivo expression patterns of genes associated with the three primary axes of diversity using transgenic reporters inserted into the endogenous loci (*Venken et al., 2011*).

Axis 1 separated clusters into two groups of four that were defined by mutually exclusive binary expression of two genes, *Fasciclin 2* (*Fas2*) and *klingon* (*klg*), respectively (*Figure 3A*), each encoding immunoglobulin (Ig) superfamily proteins. *Fas2* was expressed in LoP layers a/b, whereas *klg* was expressed in LoP layers c/d (*Figure 3B*). Clusters expressing *Fas2* and *klg* also expressed previously described markers for T4/T5 subtypes a/b (*dachshund* (*dac*)) and c/d (*bi*, *Connectin* (*Con*)) (*Apitz and Salecker, 2018*) (*Figure 3—figure supplement 1*). Thus, Axis 1 separated LoP layer a/b and c/d subtypes, defining specificity of axonal outputs between two broad domains of the LoP

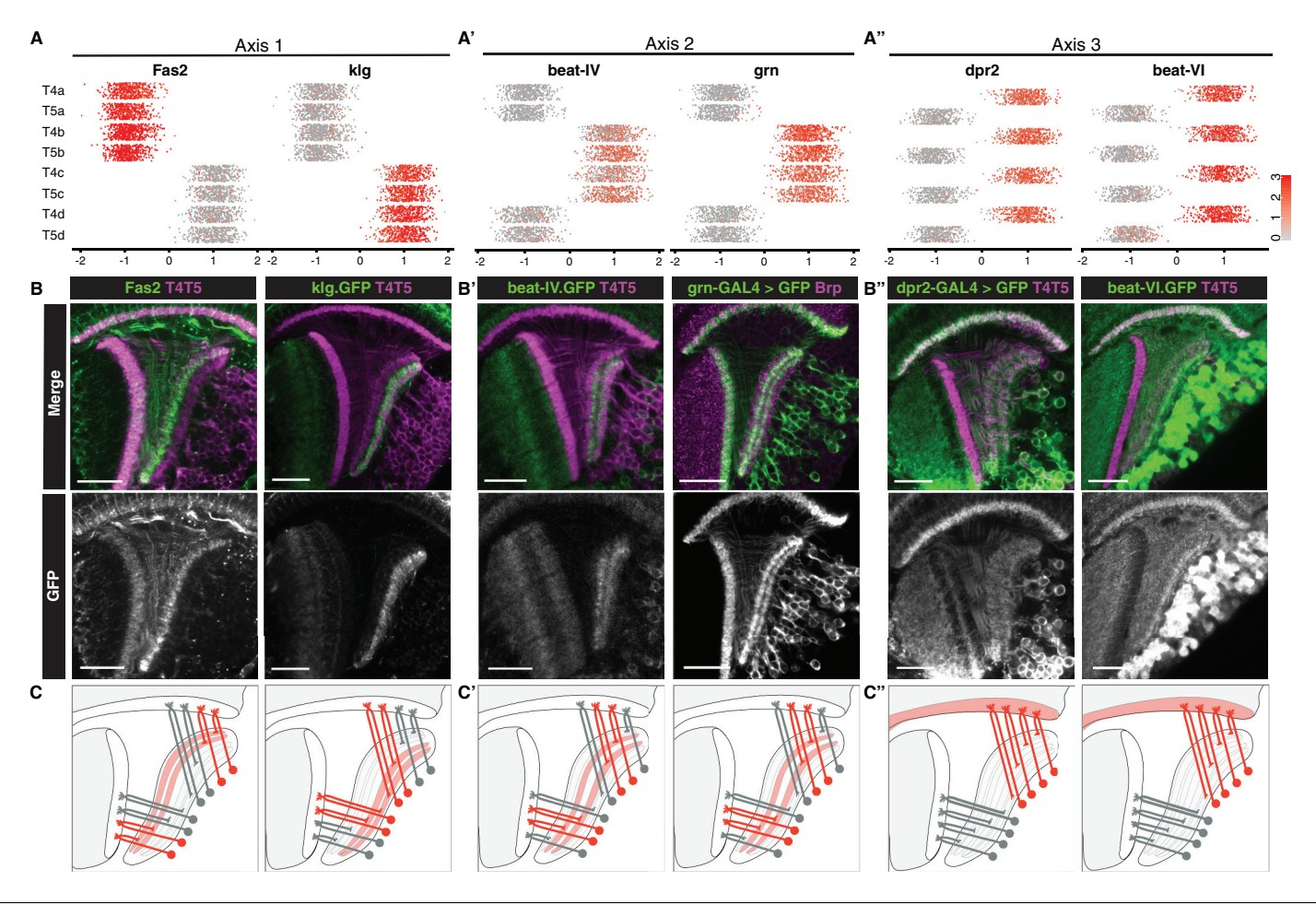

**Figure 3.** Primary axes of transcriptional diversity define groups of T4/T5 subtypes with shared wiring patterns. (A–A") 1-D scatterplots show distribution of cells along Axis 1, 2, and 3 for each cluster. Normalized expression levels are indicated by color, as in scale. (B–B") In vivo expression of marker genes for each axis at 48 hr APF. *Fas2* labels LoP layers a/b, *klg* labels LoP layers c/d, *beat-IV* and *grn* label LoP layers b/c, *dpr2* and *beat-VI* label M10 but not Lo1. Scale bars, 20 μm. Sets of positive clusters in (A) are matched to specific sets of T4/T5 subtypes based on in vivo expression patterns in (B). Individual cluster identities are deduced based on combination of expression patterns. For example, T4a is *Fas2*+ (a/b), *beat-IV*- (not b/c), *dpr2*+ (M10). (C–C") Schematic of wiring patterns of T4/T5 subtypes corresponding to the expression patterns of marker genes (red). See also *Figure 3—figure supplement 1*.

DOI: https://doi.org/10.7554/eLife.50822.005

The following figure supplement is available for figure 3:

**Figure supplement 1.** Expression patterns of known marker genes for a/b and c/d subtypes along Axis 1 at 48 hr APF.

DOI: https://doi.org/10.7554/eLife.50822.006

(*Figure 3C*). This corresponds to separation of horizontal (posterior/anterior) and vertical (upwards/downwards) motion detection circuits, respectively.

Axis 2 separated clusters into two groups of four defined by binary expression of *beat-IV* (an Ig superfamily protein) and the TF *grn*. Both genes were expressed in LoP layers b/c, but not a/d. Thus, Axis 2 further separated subtypes into inner (b and c) and outer (a and d) LoP layer subtypes in a symmetrical fashion, defining specificity of axonal outputs between adjacent layers within the two broad domains of the LoP (*Figure 3A'–3C'*). This corresponds to further separation of each of the motion detection circuits into two subcircuits detecting motion in two opposing directions (i.e. horizontal into posterior and anterior, and vertical into upwards and downwards motion).

Axis 3 separated clusters into two groups of four defined by binary expression of *dpr2* and *beat-VI*, each encoding an Ig superfamily protein. In vivo, both genes were expressed in all LoP layers, and M10 but not Lo1. Thus, Axis 3 separated all T4 from all T5 subtypes, defining specificity of

dendritic inputs (*Figure 3A''–3C''*). This corresponds to separation into two parallel circuits for ON and OFF motion detection, respectively.

Taken together, three primary axes of diversity defined distinct wiring features of T4/T5 subtypes and in combination defined wiring patterns of each T4/T5 subtype (*Figure 3A–3A''*). A combination of Axis 1 and Axis 2 defined four types of axonal outputs (a, b, c, d), and Axis 3 defined two types of dendritic inputs (T4 and T5).

## Transcriptional program of a single T4/T5 subtype

In addition to the transcriptional differences between groups of T4/T5 subtypes described above, further variation might exist at the individual subtype level. To examine this possibility, we focused on a single subtype (T4a) and performed comprehensive pairwise comparisons with each of the other subtypes (i.e. 'one versus one').

First, we compared T4a and each subtype that differed by a single wiring feature: either axonal outputs (T4b, T4c, T4d), or dendritic inputs (T5a) (*Figure 4*, upper dot plots). Comparison of T4a to T4c or T4d, which have axonal outputs in non-adjacent LoP layers, yielded the largest number of

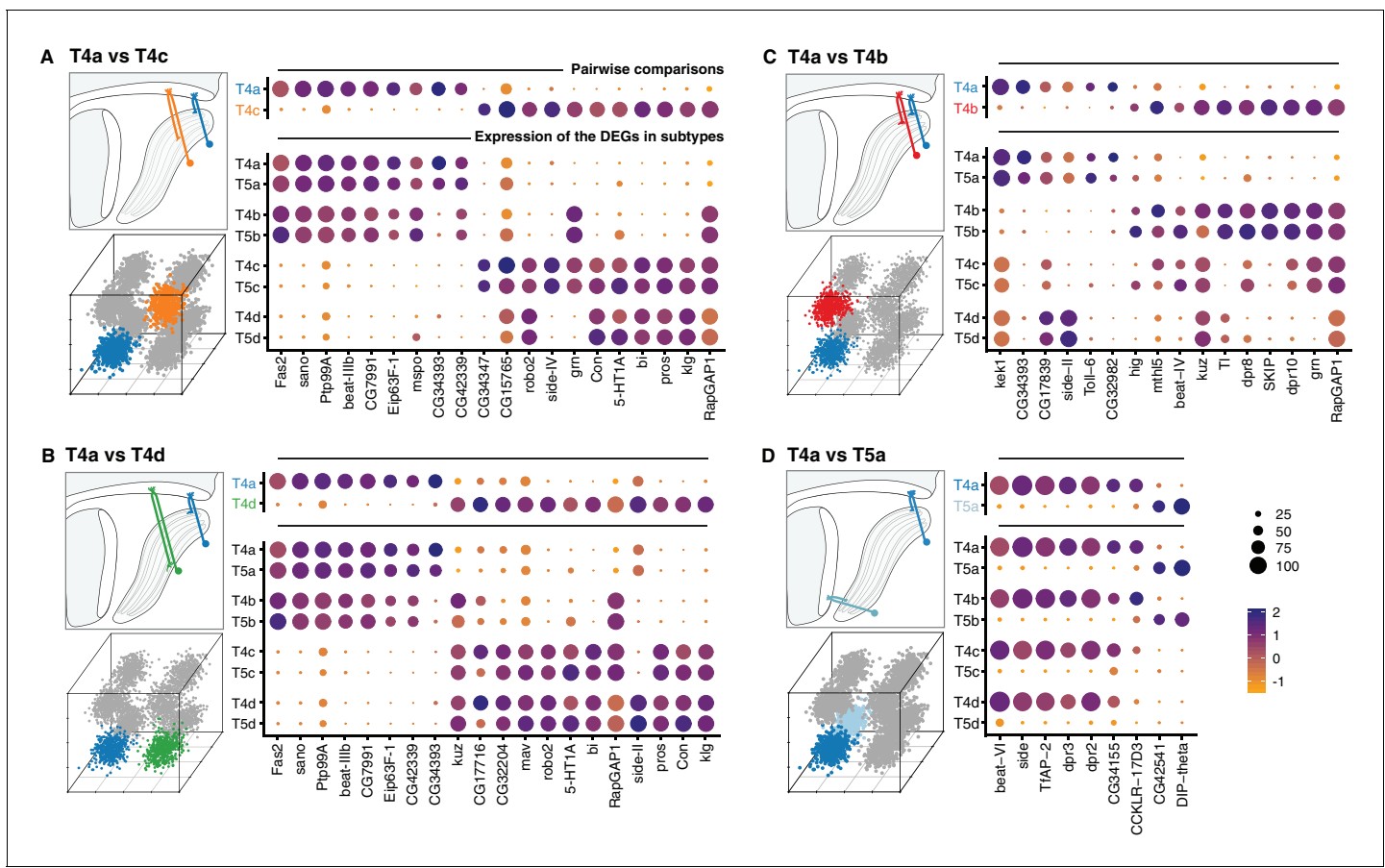

**Figure 4.** Transcriptional program of a single T4/T5 subtype. Pairwise comparisons between T4a and other subtypes ('one versus one', see Materials and methods) that differ by either axonal outputs (**A–C**), or dendritic inputs (**D**). For each comparison, insets indicate morphologies (upper left) and cluster distributions along axes of transcriptional diversity (lower left). Expression patterns of differentially expressed genes (DEGs) for each pairwise comparison are shown in upper right. Dot size indicates the percentage of cells in which the DEG was detected, color represents average scaled expression, as in scale. Genes are ordered by fold-change values. Top 20 DEGs are shown for (**A**) and (**B**). Expression patterns of DEGs among all eight subtypes are shown in lower right. See also *Figure 4—figure supplement 1*.

DOI: https://doi.org/10.7554/eLife.50822.007

The following figure supplement is available for figure 4:

**Figure supplement 1.** Pairwise comparisons between T4a and subtypes that differ by both axonal outputs and dendritic inputs.
DOI: https://doi.org/10.7554/eLife.50822.008

differentially expressed genes (DEGs, 58 and 55, respectively). T4a and T4b have axonal outputs in adjacent LoP layers, and were separated by an intermediate number of DEGs (16). Finally, only a small number of DEGs (9) separated T4a and T5a, which share axonal output but receive different dendritic inputs.

Expression patterns of the DEGs from pairwise comparisons across all T4/T5 subtypes revealed a general pattern (*Figure 4*, lower dot plots): virtually all DEGs were co-regulated across all subtypes according to either specificity of axonal outputs or dendritic inputs. In other words, distinct sets of DEGs were expressed in each pair of subtypes with shared axonal outputs, but different dendritic inputs (e.g. in T4a and T5a, *Figure 4A–C*). Similarly, a distinct set of DEGs was expressed in groups of subtypes with shared dendritic inputs, but different axonal outputs (i.e. all T4 subtypes, *Figure 4D*).

The co-regulation of DEGs according to wiring patterns was not limited to comparisons between subtypes that differed by a single wiring feature. DEGs between T4a and subtypes that differed by both axon and dendrite wiring patterns (e.g. T5c), were also expressed in groups of subtypes sharing either axonal outputs or dendritic inputs (*Figure 4—figure supplement 1*).

In addition to three primary axes of transcriptional diversity, this analysis shows that a number of DEGs exhibited more distinct LoP layer-specific patterns. For example, many DEGs were specifically expressed or suppressed in T4/T5 subtypes from a single LoP layer (*Figure 4C*).

Many of the DEGs have been implicated in neuronal wiring specificity (*Figure 4*). Approximately half of the DEGs encoded CSPs with cell adhesion domains (Ig/LRR), including multiple members of the dpr/DIP and beat/side families of interacting proteins (*Zinn and Özkan, 2017*); specific members of these families have been shown to regulate axon guidance and synaptic specificity in the developing fly nervous system.

Taken together, these results reveal that the transcriptional organization of T4/T5 neurons mirrored their wiring patterns. Discrete groups of co-regulated genes reiteratively defined either shared axon or shared dendrite wiring patterns among different subtypes. These groups of genes were assembled in different combinations to uniquely define the eight T4/T5 subtypes.

## Stable and dynamic features of T4/T5 transcriptional programs during development

To evaluate how gene expression in T4/T5 subtypes changes during development, we profiled T4/T5s at an earlier time point, 24 hr APF. Similar to our dataset at 48 hr APF, we identified eight distinct populations separated by three equivalent axes of transcriptional diversity. Many of the same genes were associated with these axes at both time points, allowing us to match subtypes between 24 hr and 48 hr APF (*Figure 5—figure supplement 1*).

Comparison of 24 hr and 48 hr datasets revealed stable and dynamic features of T4/T5 transcriptional programs. TFs defining the primary axes of diversity (*bi*, *grn*, *TfAP-2*) were expressed in the same sets of subtypes at both time points, suggesting they may contribute to stable subtype identities during development (*Figure 5A*). Some CSPs also exhibited stable expression, marking subtypes with shared axon or dendrite wiring patterns at both time points. Other CSPs were dynamically regulated and were specific to subtypes only at a particular stage of development (*Figure 5B–D* and *Figure 7—figure supplement 2*).

Interestingly, dynamic changes in gene expression were also coordinated among subtypes with shared wiring features (*Figure 5B–D*). For example, *dpr3* and a few other CSPs were synchronously upregulated in all T4 subtypes from 24 hr to 48 hr APF. Similarly, *Toll-6* was synchronously upregulated in both LoP layer 'a' subtypes (T4a/T5a). These data indicate that similar transcriptional programs unfold in parallel among T4/T5 subtypes with shared wiring features.

## Axon-specific transcriptional programs of T4/T5 neurons control lamination of LoP layers

A remarkable correspondence between transcriptional programs and wiring patterns suggested that these programs control development of corresponding features of T4/T5 neurons. During the period covered in our study (24–48 hr APF) four LoP layers form in two discrete lamination steps (*Figure 6—figure supplement 1*). The T4/T5 axon terminals first laminate into two broad domains corresponding to layers a/b and c/d. These domains then further sublaminate into two pairs of adjacent layers

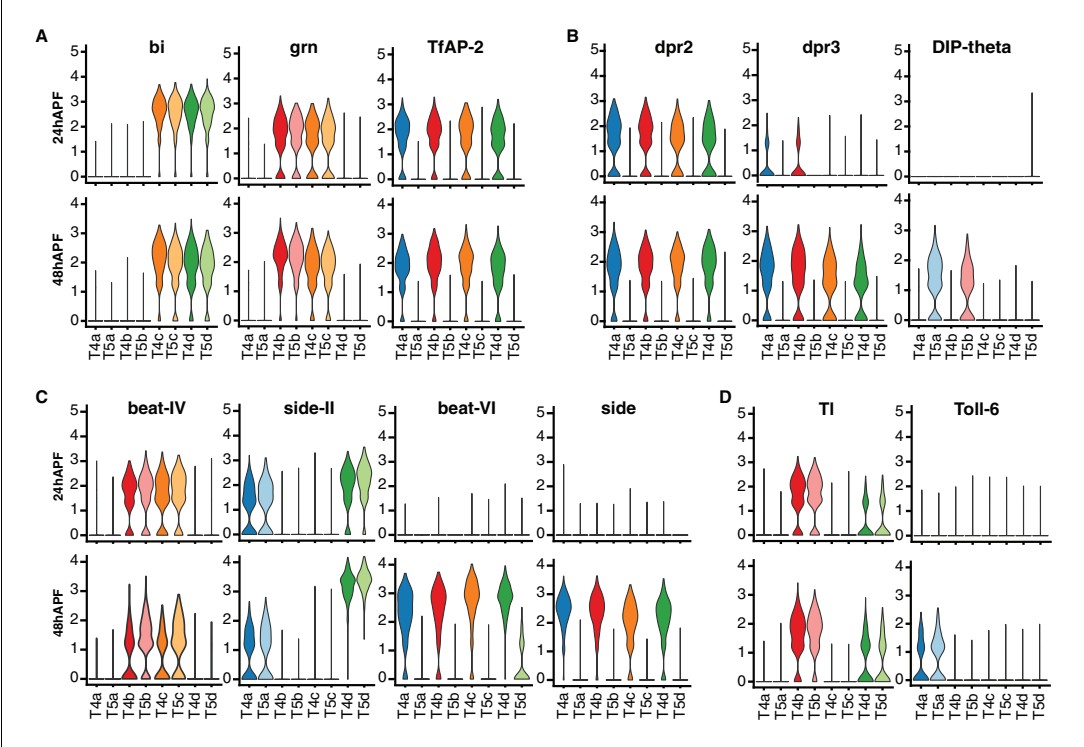

**Figure 5.** Dynamics of T4/T5 transcriptional programs during development. Distributions of normalized expression levels of TFs (**A**) and selected families of CSPs (**B-D**) at 24 hr and 48 hr APF. Distributions for each subtype are color-coded as in *Figure 1*. See also *Figure 5—figure supplement 1*.
DOI: https://doi.org/10.7554/eLife.50822.009
The following figure supplement is available for figure 5:

**Figure supplement 1.** Single-cell profiling of T4/T5 neurons at 24 hr APF.
DOI: https://doi.org/10.7554/eLife.50822.010

to form the four discrete LoP layers, a, b, c, and d. The two primary axes of transcriptional diversity mirrored these two stages of LoP layer formation, suggesting a regulatory code for axon wiring (*Figure 3*). Mutually exclusive expression of the TFs *dac* and *bi/pros* separated a/b (*dac*+) from c/d (*bi+/pros*+) subtypes (i.e. Axis 1, *Apitz and Salecker, 2018*). These subtypes were further separated by expression of the TF *grn* into inner (b and c, *grn*+) and outer (a and d, *grn*-) layer subtypes in a symmetrical fashion (i.e. Axis 2). This suggested that two levels of transcriptional regulation, acting either sequentially or in a temporally overlapping way, control development of four types of T4/T5 axonal outputs.

We sought to experimentally address this issue. Previous studies indicate that *bi* specifies c/d subtypes and formation of corresponding LoP layers. RNAi of *bi* in all T4/T5 neurons results in loss of the c/d domain of the LoP, whereas overexpression results in loss of the a/b domain. In both cases, further sublamination of remaining inner and outer LoP layer pairs still occurs (*Apitz and Salecker, 2018*). We performed RNAi of *grn*, which resulted in a different phenotype; whereas distinct a/b and c/d LoP domains were still separated by a pronounced gap and differential expression of *Con* (a marker for LoP layers c/d), both domains failed to sublaminate into inner and outer layers, instead forming a single layer each (*Figure 6A–B*). The overall morphological organization of T4/T5 neurons was otherwise unaffected. Overexpression of *grn* in all T4/T5 neurons also resulted in a specific failure of a/b and c/d LoP domains to sublaminate.

RNAi and overexpression of *grn* resulted in significant loss of T4/T5 neuron numbers between 24 hr and 48 hr APF (*Figure 6—figure supplement 2*), associated with an increase in apoptosis (*Figure 6C–D*). Failure of LoP layer sublamination could result from death of specific subtypes during development. Alternatively, differential expression of *grn* might be required to direct T4/T5 axons to discrete layers. Expression of baculovirus caspase inhibitor p35 in developing T4/T5s rescued cell

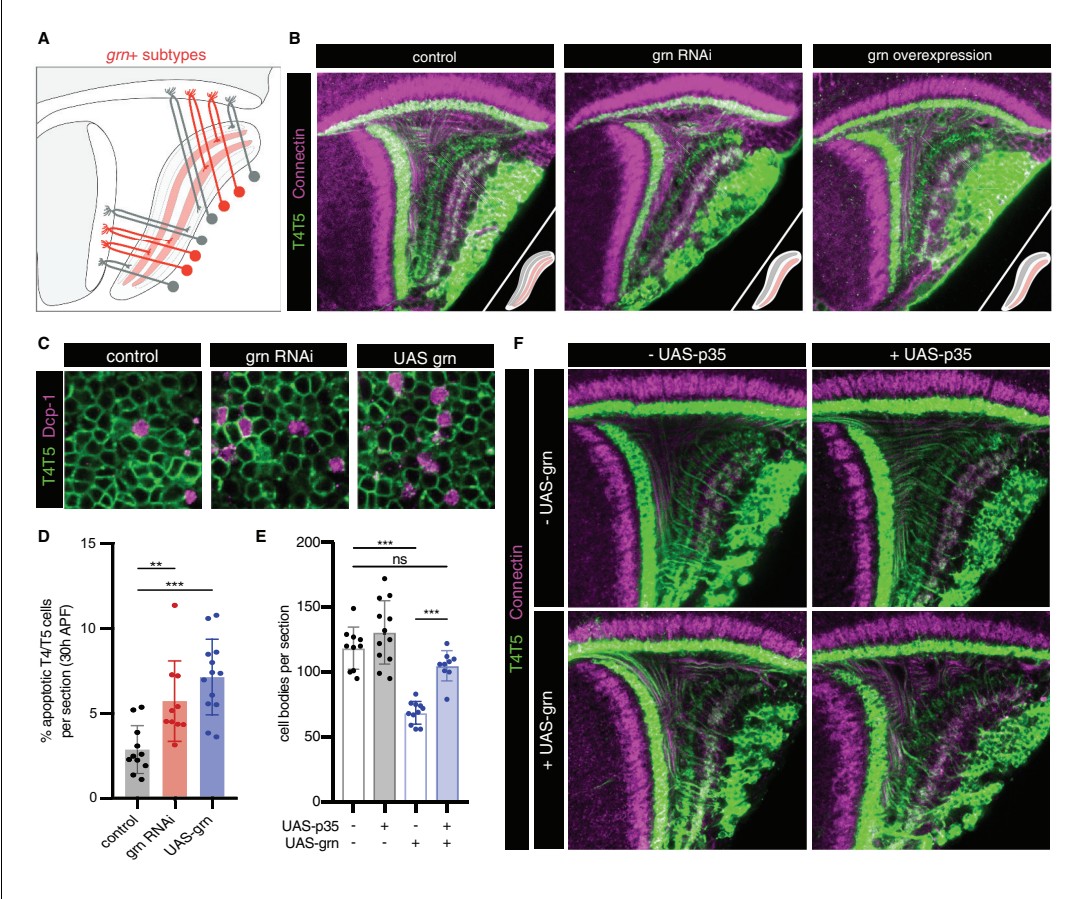

**Figure 6.** *grn* controls sublamination of T4/T5 axons into inner and outer LoP layers. (**A**) Schematic of *grn+* (red) T4/T5 subtypes in wild-type optic lobe. *grn* expression defines inner LoP layer subtypes. See also *Figure 6—figure supplement 1*. (**B**) *grn* RNAi and *grn* overexpression in all T4/T5 neurons specifically disrupts sublamination of a/b (Con-) and c/d (Con+) LoP subdomains into inner and outer layers. Insets depict LoP phenotypes. (**C–D**) Immunostaining for Death caspase-1 (Dcp-1) reveals increased apoptotic T4/T5 neurons under *grn* RNAi and overexpression (UAS-grn) conditions at 30 hr APF. See also *Figure 6—figure supplement 2*, and *Figure 6—source data 1*. (**E**) Ectopic expression of p35 in T4/T5 neurons (UAS-p35) rescues apoptotic cell death associated with overexpression of *grn*. (**F**) *grn* overexpression specifically disrupts axon sublamination when apoptosis is blocked. Statistical significance assessed by one-way ANOVA with Tukey's multiple comparison test (**p<0.01, ***p<0.001). Bars and whiskers represent mean and standard deviation. Dots represent values for individual optic lobes.

DOI: https://doi.org/10.7554/eLife.50822.011

The following source data and figure supplements are available for figure 6:

**Source data 1.** Cell number quantification data for *Figure 6D–E* and *Figure 6—figure supplement 2*.
DOI: https://doi.org/10.7554/eLife.50822.014

**Figure supplement 1.** Sequential lamination of T4/T5 axons and four LoP layers.
DOI: https://doi.org/10.7554/eLife.50822.012

**Figure supplement 2.** *grn* RNAi and *grn* overexpression cause significant loss of T4/T5 neurons between 24 and 48 hr APF.
DOI: https://doi.org/10.7554/eLife.50822.013

death associated with *grn* overexpression. Nevertheless, T4/T5 axons still failed to sublaminate into four discrete LoP layers (*Figure 6E–F*). Thus, differential expression of *grn* is specifically required for sublamination of T4/T5 axons between pairs of adjacent LoP layers.

We conclude that axon-specific transcriptional programs defined by binary (ON/OFF) expression patterns of two TFs, *bi* and *grn*, control the formation of four LoP layers and corresponding T4/T5 axonal wiring patterns.

## Discussion

Single-cell transcriptional profiling has the potential to transform our understanding of the genetic programs controlling wiring in complex nervous systems (*Li et al., 2017*; *Tasic et al., 2018*; *Klingler et al., 2018*). However, neurons exhibit a vast diversity of wiring patterns, morphologies, and molecular identities, making it difficult to extract the transcriptional logic underlying specific wiring features. Here, we turned to the closely related T4/T5 subtypes of the *Drosophila* visual system, which differ by specific variations in wiring, with the expectation that transcriptional differences among them would reflect the specificity of dendritic inputs and axonal outputs. A unique attribute of T4/T5 neurons is that the same dendritic and axonal wiring patterns are reiteratively used among different subtypes; each neuron can be described by a unique combination of one of four types of axonal outputs and one of two types of dendritic inputs. We anticipated that this property of T4/T5 neurons would provide an opportunity to assess the relationship between specific genetic programs and fundamental features of neuronal architecture.

Unsupervised analysis revealed that separable transcriptional programs correlate with these specific wiring features. We demonstrate through gain and loss of function experiments that these programs control specific axonal targeting features, which are separable from other features (e.g. dendrite targeting). These programs can be re-assembled in a modular fashion to generate neuronal subtypes with different combinations of wiring features. A modular transcriptional architecture may provide a general strategy for discrete modifications to neuronal connectivity in development and evolution.

A common T4/T5 neuronal identity is defined by a unique combination of TFs expressed in all subtypes (e.g. *Lim1*, *Drgx*, *acj6*) (*Davie et al., 2018*; *Konstantinides et al., 2018*). Perturbation of TFs expressed in all subtypes disrupts overall organization of T4/T5 neurons, including both dendritic and axonal morphologies (*Contreras et al., 2018*; *Schilling et al., 2019*). We find that this common T4/T5 transcriptional program is further diversified by separable feature-specific transcriptional programs. These programs are defined by three binary (ON/OFF) TF expression patterns, with two TF patterns defining the four axonal outputs and one TF pattern defining the two dendritic inputs. In this way, modular TF codes defining common and feature-specific transcriptional programs give rise to eight T4/T5 subtypes (*Figure 7*).

Four pairs of T4/T5 subtypes with shared axonal outputs (and different dendritic inputs) each target one of four LoP layers, a-d. The ultimate layered architecture of neuropils develops through sequential lamination into increasing numbers of layers (*Sanes and Zipursky, 2010*; *Millard and Pecot, 2018*). Together with previous results, our findings suggest that the lamination of T4/T5 axonal outputs occurs via two distinct processes, each controlled by a separate TF. Binary expression of *bi* is required for lamination of the broad a/b from c/d LoP domains (*Apitz and Salecker, 2018*), whereas binary expression of *grn* is required for sublamination of each of these two domains into separate LoP layers. Importantly, perturbation of each TF exclusively disrupts the corresponding lamination step, while not affecting other morphological features of T4/T5 neurons. Similarly, two quartets of subtypes with shared dendritic inputs (and different axonal outputs) were defined by binary expression of *TfAP-2*. Arborization of dendrites in M10 (T4) or Lo1 (T5) occurs during initial neurite guidance steps, preceding the developmental stages covered in this study (*Pinto-Teixeira et al., 2018*). We hypothesize that DEGs between T4 and T5 subtypes identified in our analysis contribute to the connections with two distinct sets of presynaptic partners (*Shinomiya et al., 2019*).

The binary expression patterns of TFs also mirror the developmental lineages of T4/T5 neurons. a/b and c/d subtypes arise from *bi*- and *bi*+ progenitor populations. Neuroblasts from each population undergo two terminal Notch-dependent asymmetric divisions to give rise to the eight subtypes (*Pinto-Teixeira et al., 2018*). These divisions correspond to binary expression patterns of *grn* and *TfAP-2*, respectively, which act with Notch signaling to regulate wiring. Remarkably, despite divergent developmental trajectories separated by multiple divisions and distinct progenitor pools, all T4 and all T5 subtypes converge onto the same transcriptional programs associated with two types of dendritic inputs. Three regulatory dichotomies could also reflect the evolutionary origin of T4/T5 subtypes and correspond to consecutive duplications of ancestral cell types and circuits (*Shinomiya et al., 2015*; *Arendt et al., 2016*).

Each axonal and dendritic transcriptional program is characterized by a specific pattern of TFs, as well as a set of CSPs, many of which are implicated in regulating wiring in other developmental

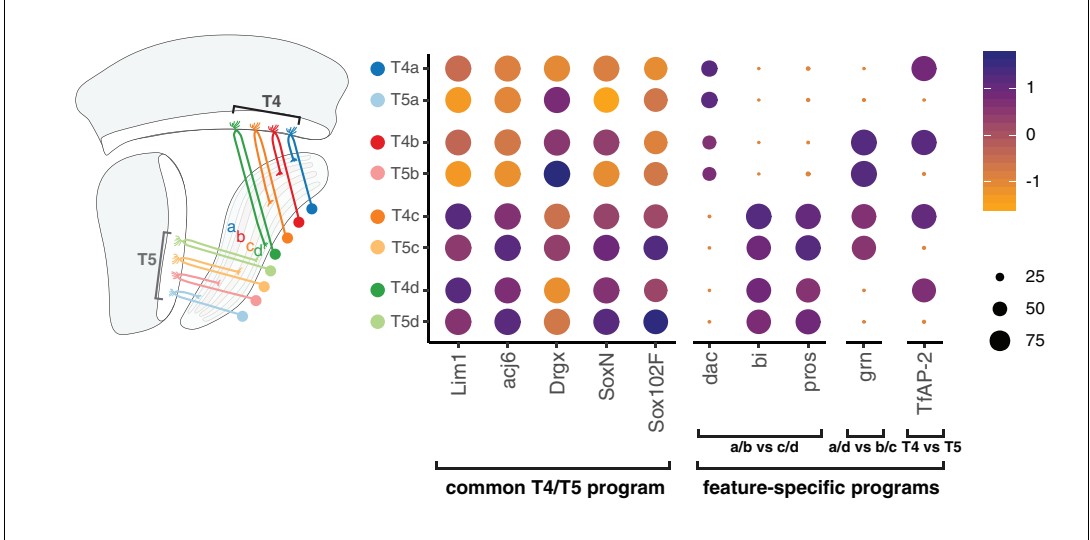

**Figure 7.** Modular transcription factor codes define eight T4/T5 subtypes. A common T4/T5 regulatory program is defined by TFs expressed in all subtypes (*Davie et al., 2018*; *Konstantinides et al., 2018*; *Contreras et al., 2018*; *Schilling et al., 2019*). This program is diversified by modular combinations of feature-specific TFs defining unique wiring patterns of eight T4/T5 subtypes. Dot size indicates the percentage of cells in which the TF was detected, color represents average scaled expression, as in scale. Data shown for 48 hr APF. See also *Figure 7—figure supplements 1* and *2*.
DOI: https://doi.org/10.7554/eLife.50822.015

The following figure supplements are available for figure 7:

**Figure supplement 1.** Expression patterns of TFs at 24 hr APF.
DOI: https://doi.org/10.7554/eLife.50822.016
**Figure supplement 2.** Expression patterns of subtype-enriched CSPs with cell adhesion domains.
DOI: https://doi.org/10.7554/eLife.50822.017

contexts. These include Ig superfamily proteins in which different paralogs exhibit discrete heterophilic binding specificities, including the beat/side and the dpr/DIP interacting protein families (*Zinn and Özkan, 2017*). Interestingly, dynamic expression of these proteins in neurons with shared wiring features was developmentally coordinated. We envision that the synaptic specificity of T4/T5 dendrites and axons are determined by the combined activity of these recognition molecules through interactions with synaptic partners. Future experiments utilizing gain and loss of function analysis, either alone or different combinations, will provide insights into the cellular recognition mechanisms by which synaptic specificity is established.

The composite morphological properties of T4/T5 subtypes allowed us to identify, and thus decouple transcriptional programs for dendrite and axon wiring. Combining separate dendritic and axonal programs, and variations on them, may contribute to the diversification of synaptic specificity in different neuronal subtypes across complex nervous systems

## Materials and methods

### Key resources table

| Reagent type (species) or resource | Designation | Source or reference | Identifiers | Additional information |
|---|---|---|---|---|
| Genetic reagent (*D. melanogaster*) | MCFO-1 (pBPhsFLP2::PEST;+; UAS-FSF-smGdP::HA_V5_FLAG) | PMID: 25964354 | RRID: BDSC_64085 | Gift from Aljoscha Nern and Gerald Rubin |
| Genetic reagent (*D. melanogaster*) | 10XUAS-IVS-myr::tdTomato | Bloomington Drosophila Stock Center | RRID: BDSC_32222 | |
| Genetic reagent (*D. melanogaster*) | 23G12-GAL4 (T4/T5) | Bloomington Drosophila Stock Center | RRID: BDSC_49044 | T4/T5 driver |

*Continued on next page*

*Continued*

| Reagent type (species) or resource | Designation | Source or reference | Identifiers | Additional information |
|---|---|---|---|---|
| Genetic reagent (*D. melanogaster*) | 42F06-GAL4 (T4/T5) | Bloomington Drosophila Stock Center | RRID: BDSC_41253 | T4/T5 driver |
| Genetic reagent (*D. melanogaster*) | 23G12-LexA (T4/T5) | Bloomington Drosophila Stock Center | RRID: BDSC_65044 | T4/T5 driver |
| Genetic reagent (*D. melanogaster*) | {R59E08-p65ADZp (attP40); R42F06-ZpGdbd (attP2)} (T4/T5 splitGAL4) | PMID: 28384470 | JRC_SS00324 | |
| Genetic reagent (*D. melanogaster*) | UAS-CD4-tdGFP | Bloomington Drosophila Stock Center | RRID: BDSC_35839 | |
| Genetic reagent (*D. melanogaster*) | UAS-H2A::GFP | PMID: 26687360 | N/A | Gift from Barret Pfeiffer and Gerald Rubin |
| Genetic reagent (*D. melanogaster*) | LexAop-myr::tdTomato | PMID: 24462095 | N/A | |
| Genetic reagent (*D. melanogaster*) | 10XUAS-IVS-myr::GFP | Bloomington Drosophila Stock Center | RRID: BDSC_32197 | |
| Genetic reagent (*D. melanogaster*) | 10XUAS-IVS-mCD8::RFP | Bloomington Drosophila Stock Center | RRID: BDSC_32219 | |
| Genetic reagent (*D. melanogaster*) | Mi{PT-GFSTF.1}klg [MI02135-GFSTF.1] | Bloomington Drosophila Stock Center | RRID: BDSC_59787 | |
| Genetic reagent (*D. melanogaster*) | Mi{PT-GFSTF.1}beat-IV [MI05715-GFSTF.1] | Bloomington Drosophila Stock Center | RRID: BDSC_66506 | |
| Genetic reagent (*D. melanogaster*) | dpr2-Gal4 | Hugo J. Bellen | N/A | Gift from Hugo J. Bellen |
| Genetic reagent (*D. melanogaster*) | P{w[+mW.hs]=GawB} grn[05930-GAL4] | Bloomington Drosophila Stock Center | RRID: BDSC_42224 | |
| Genetic reagent (*D. melanogaster*) | Mi{y[+mDint2]=MIC} beat-VI[MI13252] | Bloomington Drosophila Stock Center | RRID: BDSC_58680 | |
| Genetic reagent (*D. melanogaster*) | P{y[+t7.7] v[+t1.8]=TRiP. HMS01085}attP2 | Bloomington Drosophila Stock Center | RRID: BDSC_33746 | UAS-grnRNAi |
| Genetic reagent (*D. melanogaster*) | P{UAS-p35.H}BH1 | Bloomington Drosophila Stock Center | RRID: BDSC_5072 | |
| Genetic reagent (*D. melanogaster*) | UAS-grn.ORF.3xHA | FlyORF | Stock #: F001916 | |
| Antibody | Chicken polyclonal anti-GFP | Abcam | Cat. #: ab13970 RRID: AB_300798 | IHC (1:1000) |
| Antibody | Rabbit polyclonal anti-dsRed | Clontech | Cat. #: 632496 RRID: AB_10013483 | IHC (1:200) |
| Antibody | Mouse monoclonal anti-Brp | Developmental Studies Hybridoma Bank | Cat. #: nc82 RRID: AB_2314866 | IHC (1:20) |
| Antibody | Mouse monoclonal anti-V5 | Abcam | Cat. #: ab27671 RRID: AB_471093 | IHC (1:300) |
| Antibody | Rabbit polyclonal anti-Dcp-1 | Cell Signalling | Cat. #: 9578 RRID: AB_2721060 | IHC (1:50) |
| Antibody | Goat polyclonal anti-chicken IgY Alexa Fluor 488 | Invitrogen | Cat. #: A11039 RRID: AB_142924 | IHC (1:200) |
| Antibody | Goat polyclonal anti-mouse IgG Alexa Fluor 488 | Invitrogen | Cat. #: A11029 RRID: AB_138404 | IHC (1:500) |
| Antibody | Goat monoclonal anti-rabbit IgG Alexa Fluor 568 | Invitrogen | Cat. #: A11011 RRID: AB_143157 | IHC (1:200) |

*Continued*

| Reagent type (species) or resource | Designation | Source or reference | Identifiers | Additional information |
|---|---|---|---|---|
| Antibody | Goat polyclonal anti-rat IgG Alexa Fluor 568 | Invitrogen | Cat. #: A11077 RRID: AB_141874 | IHC (1:500) |
| Antibody | Goat oligoclonal anti-rabbit IgG Alexa Fluor 647 | Invitrogen | Cat. #: A27040 RRID: AB_2536101 | IHC (1:200) |
| Antibody | Donkey polyclonal anti-mouse IgG Cy5 | Jackson Immuno Research Laboratories | Cat. #: 715-175-150 RRID: AB_2340819 | IHC (1:200) |
| Chemical compound, drug | Papain | Worthington | Cat. #: LK003178 | |
| Chemical compound, drug | Liberase protease | Sigma-Aldrich | Cat. #: 5401119001 | |
| Software, algorithm | Cell Ranger 2.2.0 | https://10xgenomics.com | RRID:SCR_017344 | |
| Software, algorithm | Seurat 2.3.4 | https://satijalab.org/seurat/ | RRID: SCR_016341 | |

## Animal husbandry

Flies (*Drosophila melanogaster*) were reared at 25°C on standard medium. For developmental analysis by immunohistochemistry, sorting, and sequencing, white pre-pupae (0 hr APF) were collected and incubated for indicated number of hours.

## Fly stocks

Multiple-transgene genotypes are enclosed in brackets. The following transgenic lines were used in this study: MCFO-1 {pBPhsFLP2::PEST;+; UAS-FSF-smGdP::HA_V5_FLAG} (gift from Aljoscha Nern and Gerald Rubin), 10XUAS-IVS-myr::tdTomato (Bloomington Drosophila Stock Center (BDSC #32222)), 23G12-Gal4 (BDSC #49044), {R59E08-p65ADZp (attP40); R42F06-ZpGdbd (attP2)} (JRC_SS00324, Aljoscha Nern and Gerald Rubin), UAS-H2A::GFP (Barret Pfeiffer and Gerald Rubin), UAS-CD4-tdGFP (BDSC #35839), 23G12-LexA (BDSC #65044), LexAop-myr::tdTomato (Zipursky laboratory), 10XUAS-myr::GFP (Zipursky laboratory), 10XUAS-IVS-mCD8::RFP (BDSC #32219), Mi{PT-GFSTF.1}klg[MI02135-GFSTF.1] (BDSC #59787), Mi{PT-GFSTF.1}beat-IV[MI05715-GFSTF.1] (BDSC #66506), dpr2-Gal4 (Zipursky laboratory), P{w[+mW.hs]=GawB}grn[05930-GAL4] (BDSC #42224), Mi{y[+mDint2]=MIC}beat-VI[MI13252] (BDSC #58680), P{y[+t7.7] v[+t1.8]=TRiP.HMS01085}attP2 (UAS-grn-RNAi) BDSC #33746), UAS-grn.ORF.3xHA (FlyORF #F001916), 42F06-Gal4 (BDSC #41253), UAS-p35 (BDSC #5072).

For visualization of T4a clone, virgin females {pBPhsFlp2::PEST; 10XUAS-IVS-myr::tdTomato; UAS-FSF-smGdP::HA_V5_FLAG/CyO::TM6B} were crossed to males with the T4/T5-specific Split-Gal4 driver {R59E08-p65ADZp (attP40); R42F06-ZpGdbd (attP2)} (JRC_SS00324). White pre-pupae were heat shocked at 37°C for 3 min. For FACS sorting of GFP+ T4/T5 neurons, 23G12-Gal4 was used to drive UAS-H2A::GFP. For T4/T5 developmental timecourse, 23G12-Gal4 was used to drive UAS-H2A::GFP and 10XUAS-IVS-mCD8:RFP. For visualization of all T4/T5 neurons with subtype-specific markers, female virgins of the genotypes: {23G12-LexA; LexAop-myr::tdTomato; 10XUAS-myr::GFP/CyO::TM6B} or {w; 10xUAS-IVS-mCD8::RFP; 23G12-Gal4} were crossed to males with MiMICs or their derivatives for *klg*, *beat-IV*, *dpr2*, *beat-VI*, or to w[1] males for *Fas2* immunolabeling. For visualization of *grn*-expressing neurons, grn-Gal4 was used to drive 10XUAS-myr::GFP. For *grn* phenotypes, virgin females {w[1];UAS-CD4-tdGFP;23G12-Gal4} were crossed to males with UAS-grn-RNAi or UAS-grn.ORF.3xHA transgenes. For p35 rescue experiments, virgin females with 23G12-Gal4, UAS-CD4-tdGFP and with or without UAS-grn.ORF.3xHA were crossed to males with 42F06-Gal4, and with or without UAS-p35 transgene, as indicated. All RNAi and overexpression crosses were raised at 29°C.

## Immunohistochemistry/Immunofluorescence

Brains were dissected in ice-cold Schneider's *Drosophila* Medium (Gibco #21720–024), and fixed in PBS (Bioland Scientific LLC #PBS01-03) containing 4% paraformaldehyde (Electron Microscopy

Sciences, Cat#15710) for 25 min at room temperature (RT). Brains were rinsed repeatedly with PBST (PBS containing 0.5% Triton-X100 (Sigma #T9284)), and incubated in blocking solution (PBST containing 10% Normal Goat Serum (Sigma #G6767)) for at least 1 hr at RT prior to incubation with antibody. Brains were incubated sequentially with primary and secondary antibodies diluted in blocking solution overnight at 4C, with at least 2 PBST rinses followed by 2 hr incubations at RT in between and afterwards. Brains were transferred to 50% (for 30 min), then 100% EverBrite mounting medium (Biotium #23001) and mounted on slides for confocal microscopy.

Primary antibodies and dilutions used in this study were as follows: chicken anti-GFP (abcam #13970, 1:1000), rabbit anti-dsRed (Clontech #632496, 1:200), mouse anti-Brp (nc82 from Developmental Studies Hybridoma Bank (DSHB), 1:30), mouse anti-Fasciclin II (1D4 from DSHB, 1:20), mouse anti-V5 (abcam #ab27671, 1:300), rabbit anti-Dcp-1 (Cell Signaling Technology #9578, 1:50). Secondary antibodies and dilutions used in this study were as follows: goat anti-chicken Alexa Fluor 488 (AF488) (Invitrogen #A11039, 1:200), goat anti-mouse AF488 (Invitrogen #A11029, 1:500), goat anti-rabbit AF568 (Invitrogen #A11011, 1:200), goat anti-rat 568 (Invitrogen #A11077, 1:500), goat anti-rabbit AF647 (Invitrogen #A27040, 1:200), and donkey anti-mouse Cy5 (Jackson ImmunoResearch #715-175-150, 1:200).

## Confocal microscopy and image analysis

Immunofluorescence images were acquired using a Zeiss LSM 880 confocal microscope with Zen digital imaging software. Optical sections or maximum intensity projections were level-adjusted, cropped and exported for presentation using Image J software (Fiji). Reported expression patterns were reproducible across three or more biological samples. For cell number quantifications, optic lobes were mounted with ventral side facing objective (as in *Figure 1*), and a single optical section per lobe was acquired at 3/8 total depth in z-dimension through M10. For quantification of apoptosis, optic lobes were mounted with posterior side facing objective, and a superficial optical section with approximately 300 T4/T5 cell bodies was acquired per lobe. The section depth was determined with Dcp-1 immunofluorescence channel turned off. Files were randomized, and cell numbers and proportion of apoptotic cells were quantified blind to condition using Fiji.

## Single-cell transcriptome profiling

### Purification of genetically labelled T4/T5 neurons

Males with 23G12-Gal4 driver were crossed to virgin females with UAS-H2A::GFP reporter. F1-generation female white pre-pupae were collected at 0 hr APF and reared at 25˚C. Optic lobes were dissected out at 24 hr and 48 hr APF from 27 and 18 pupae, respectively. Brain tissue was incubated in papain (Worthington #LK003178) and Liberase protease (Sigma-Aldrich #5401119001) cocktail at 25˚C for 15 min. Next, tissue was gently washed twice with 1X PBS and dissociated mechanically by pipetting. Cell suspension was filtered through 20 µm cell-strainer (Corning #352235). Single-cell suspension was FACS sorted (BD FACSAria II) to isolate GFP-positive cells.

### Single-cell library preparation and sequencing

FACS-sorted single-cells were captured from a cell suspension using the 10X Chromium platform (~6500–7000 cells loaded). Single-cell RNA-Seq libraries were generated using Chromium Single Cell Reagent Kit V2 according to the manufacturer's protocol, with 12 cycles of PCR for cDNA amplification. RNA-Seq libraries were sequenced using Illumina Hiseq 4000 platform (paired-end 100 bp reads). Each sample was captured and sequenced using one lane of 10X Chromium and one lane of HiSeq 4000.

## Raw data processing

Raw Illumina base call files (*.bcl files) were converted into fastq files using bcl2fastq (–use-bases-mask=Y26 n*,I8n*,Y100n*). Fastq files were processed using Cell Ranger (2.2.0) pipeline with default parameters. Reference transcriptome package for Cell Ranger was generated using *Drosophila melanogaster* genome sequence and gene annotations from FlyBase (release 6.22). Both samples were sequenced at mean depth of 92,000 reads per cell (92% saturation). Average fractions of reads uniquely (confidently) mapped to genome and transcriptome were 93% and 83%, respectively.

**Single-cell data analysis**

All steps of single-cell data analysis were performed using functions and methods implemented in Seurat package (2.3.4) (*Butler et al., 2018*). Analysis for 24 hr and 48 hr datasets were performed separately.

## Quality control and data pre-processing

For 24 hr dataset, we recovered 3833 cells (median of 1447 genes and 3353 transcripts per cell). For 48 hr dataset, we recovered 3894 cells (median of 1633 genes and 4389 transcripts per cell). Initial set of cells was pre-filtered based on total number of detected genes (min. 1000; max. 2000), and percentage of mitochondrial transcripts (max. 5%). After pre-filtering, 3312 and 3620 cells remained for 24 hr and 48 hr datasets, respectively. Raw transcript counts were log-normalized using NormalizeData function. Next, we regressed out total number of transcripts per cell (nUMI) and scaled expression values to Z-scores using ScaleData function.

## Preliminary dimensionality reduction and detection of outlier cells

Sets of highly variable genes were selected using FindVariableGenes function (x.low.cutoff: 0.1, x.high.cutoff: 5, y.cutoff: 0.5). Highly variable genes were used to perform independent component analysis (ICA) using RunICA function. Independent components (ICs) were manually inspected to identify and flag outlier cells. In total, 241 and 60 cells were flagged as outliers in 24 hr and 48 hr datasets, respectively. Outlier cells were removed from subsequent steps of the analysis. After quality control and filtering, final datasets included 3071 cells for 24 hr, and 3557 cells for 48 hr datasets.

## Dimensionality reduction and clustering (48 hr APF)

We repeated selection of highly variable genes on final datasets using the same parameters (2290 genes), and used them to perform ICA. Inspection of results of ICA revealed that the three ICs separated cells into two discrete populations of approximate halves. Final clustering was performed based on these 3 ICs using the graph based clustering approach implemented in FindClusters function with default parameters. In addition to ICA, we performed principal component analysis (PCA) using the same set of highly variable genes. Comparison of ICA and PCA results revealed robustness of clusters identified by both methods (*Figure 1—figure supplement 1*).

t-distributed stochastic neighbor embedding (tSNE) was used to visualize cellular heterogeneity based on ICA and PCA results using RunTSNE function (perplexity: 100). Clusters were validated and matched to eight morphological T4/T5 subtypes using in vivo expression patterns of marker genes (*Figure 3*).

## Dimensionality reduction and clustering (24 hr APF)

Similar to 48 hr dataset, we selected highly variable genes (2198 genes), and used them to perform ICA. We selected three ICs that were driven by similar sets of genes as ICs used for clustering of 48 hr dataset. Selected ICs were used to perform clustering and tSNE using same parameters as 48 hr dataset. Cluster identities were matched with T4/T5 subtypes using expression patterns of the same sets of marker genes (*Figure 5—figure supplement 1*). In comparison to 48 hr dataset, differences between subtypes at 24 hr were less pronounced. This may reflect a lower degree of transcriptional divergence among distinct subtypes at earlier stages of development.

## Differential gene expression analysis

Differentially expressed genes (DEGs) were identified using Wilcoxon rank-sum test implemented in FindMarkers function (min.pct: 0.25, min.diff.pct: 0.25; fold-change: 1.5). We performed this analysis for each cluster against all other cells in dataset ('one versus all', *Figure 1*), and between pairs of individual clusters ('one versus one', *Figure 4* and *Figure 4—figure supplement 1*).

**Data availability**

Raw sequencing data (fastq-files), single-cell expression matrix and cell clustering results were deposited to NCBI Gene Expression Omnibus (GEO) under accession: GSE126139.

## Acknowledgements

We thank members of the Zipursky lab, Joshua Sanes, Karthik Shekhar, and Jonathan Flint for critical discussion of the manuscript. We thank Gerald Rubin, Hugo J Bellen, Aljosha Nern, Orkun Akin, and Alain Garces for transgenic fly lines and antibodies. We thank Donghui Cheng and Owen Witte for assistance in FACS purification of cells, and UCLA TCGB and BSCRC BioSequencing Core Facility for assistance with single-cell RNA-Sequencing. This work was supported by the NIH National Institute of Neurological Disorders and Stroke (T32NS048004) (SAL), and the G Harold and Leila Y Mathers Foundation (SLZ). SLZ is an Investigator of the Howard Hughes Medical Institute.

## Additional information

### Funding

| Funder | Grant reference number | Author |
|---|---|---|
| Howard Hughes Medical Institute | | S Lawrence Zipursky |
| G Harold and Leila Y. Mathers Foundation | | S Lawrence Zipursky |
| National Institute of Neurological Disorders and Stroke | T32NS048004 | Samuel A LoCascio |

The funders had no role in study design, data collection and interpretation, or the decision to submit the work for publication.

### Author contributions

Yerbol Z Kurmangaliyev, Conceptualization, Data curation, Formal analysis, Investigation, Writing—original draft, Generated single-cell sequencing data and performed single-cell data analysis; Juyoun Yoo, Conceptualization, Validation, Investigation, Writing—original draft, Generated single-cell sequencing data and performed expression validation experiments; Samuel A LoCascio, Conceptualization, Validation, Investigation, Writing—original draft, Performed expression validation experiments and functional experiments; S Lawrence Zipursky, Conceptualization, Supervision, Investigation, Writing—original draft

### Author ORCIDs

Yerbol Z Kurmangaliyev (iD) https://orcid.org/0000-0003-4829-2025
S Lawrence Zipursky (iD) https://orcid.org/0000-0001-5630-7181

### Decision letter and Author response

Decision letter https://doi.org/10.7554/eLife.50822.022
Author response https://doi.org/10.7554/eLife.50822.023

## Additional files

### Supplementary files

• Transparent reporting form  DOI: https://doi.org/10.7554/eLife.50822.018

### Data availability

Raw sequencing data, single-cell expression matrix and cell clustering results were deposited to NCBI GEO under accession: GSE126139.

The following dataset was generated:

| Author(s) | Year | Dataset title | Dataset URL | Database and Identifier |
|---|---|---|---|---|
| Kurmangalyev YZ, Yoo J, LoCascio SA, Zipursky SL | 2019 | Modular transcriptional programs separately define axon and dendrite connectivity | https://www.ncbi.nlm.nih.gov/geo/query/acc.cgi?acc=GSE126139 | NCBI Gene Expression Omnibus, GSE126139 |

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
