## [Decision Letter]

**Acceptance summary:**

This manuscript addresses the central issue of how transcription factor (TF) codes dictate neuronal sub-type cell fate, with particular emphasis upon how the proper mix of cell surface proteins (CSPs) are activated in each sub-type. Ultimately, this work informs regarding the genetic logic underlying the regulation of neuronal connectivity, addressed using single-cell sequencing of a group of eight closely related neurons with different wiring specificity.

**Decision letter after peer review:**

Thank you for submitting your article "Modular transcriptional programs separately define axon and dendrite connectivity" for consideration by *eLife*. Your article has been reviewed by 2 peer reviewers, and the evaluation has been overseen by a Reviewing Editor and Eve Marder as the Senior Editor. The following individuals involved in review of your submission have agreed to reveal their identity: Bing Ye (Reviewer #1).

The reviewers have discussed the reviews with one another and the Reviewing Editor has drafted this decision to help you prepare a revised submission. Please aim to submit the revised version within two months.

Summary:

This manuscript addresses the central issue of how transcription factor (TF) codes dictate neuronal sub-type cell fate, with particular emphasis upon how the proper mix of cell surface proteins (CSPs) are activated in each sub-type. They focus upon the T4/T5 neurons of the *Drosophila* visual system, and perform scRNA-seq upon FACS-purified cells. They find that T4/T5 neurons belong to eight major sub-types, and identify TF and CSP signatures for these groups. They functionally address the role of one TF, *grn*, and find that it controls a particular feature of the sub-types expressing this TF. Another TF with sub-type defining expression, bi, was addressed in a previously published study. The authors propose a model whereby TF codes, rather than single TFs, govern CSP codes, to thereby govern neuronal morphology. This general notion has been around for quite some time, but detailed examples of precise players and pathways have been frustratingly slow to emerge. Hence, progress in this area is of great importance to the field of developmental neurobiology.

Major Concerns:

1) It would be valuable to separately show what the TF and CSP codes are for each of the eight sub-types. More specifically, it is unclear to this reviewer what the TF codes are for the eight sub-types of T4/T5 neurons. They discuss bi, *grn* and TfAP-2, and briefly mention some others, but what are the complete TFs codes for each sub-type? Most of the data depict a mix of DEG, CSPs and TFs. Can the authors address this issue in greater depth?

Minor Concerns:

1) Can the authors also conduct RNAi of TfAP-2 and/or other TFs that they believe may specify T4/T5 sub-types to confirm some aspect of functional significance. This would be a substantive addition to the manuscript, elevating significance and impact, and they authors may likely have some of this data already in place?

2) It is difficult to understand if the *grn* LOF and GOF phenotype, and the previously published bi phenotype, fits with a combinatorial model for T4/T5 sub-type cell fate specification. Could the authors discuss this issue?

4) Figure 6: They use UAS-p35 to suppress the apoptosis triggered by *grn* overexpression, and could observe a similar phenotype even with apoptosis suppressed. Can they use the same approach to suppress apoptosis in grn RNAi, to see if this phenotype is also independent of apoptosis?

5) When describing the T4/T5 neurons maybe they could elaborate more regarding the numbers of these neurons, and previous predictions of sub-types, based upon gene expression and morphology.

6) How does the 24h APF and 48h APF time-point used for scRNA-seq relate to when the T4/T5 neurons are born?

---

## [Author Response]

Major Concerns:1) It would be valuable to separately show what the TF and CSP codes are for each of the eight sub-types. More specifically, it is unclear to this reviewer what the TF codes are for the eight sub-types of T4/T5 neurons. They discuss bi, grn and TfAP-2, and briefly mention some others, but what are the complete TFs codes for each sub-type? Most of the data depict a mix of DEG, CSPs and TFs. Can the authors address this issue in greater depth?

We agree that the textual summary of our findings in the Discussion does not do a good job of describing the modular organization of T4/T5 transcriptional programs we have uncovered in this analysis (Discussion, third paragraph). We have expanded the description and most importantly included three new figures (Figure 7 and Figure 7—figure supplements 1 and 2) to emphasize this modularity. We also include expression patterns of TFs that were shown to distinguish T4/T5s from other neurons in *Drosophila* brain (Davie et al., 2018; Konstantinides et al., 2018), or have been shown to regulate common T4/T5 morphology (Contreras et al., 2018; Schilling et al., 2019). We show that these TFs represent a common regulatory program for all T4/T5 neurons. Modular assembly of feature-specific TF patterns overlying this common T4/T5 program generates each T4/T5 subtype. These TF codes are stable and define each subtype at both 24h and 48h APF (Figure 7 and Figure 7—figure supplement 1).

CSP patterns expressed by each subtype are more complex and dynamic than the pattern of transcription factors. We now include a supplementary figure with summary of expression patterns of all subtype-enriched CSPs with cell adhesion domains at both 24h and 48h APF (Figure 7—figure supplement 2). As with the TFs, the combinations of CSPs are also largely expressed in modular fashion across subtypes with shared wiring features. While the contents of Figure 7 and its supplementary figures partially overlap with data presented in other figures, we believe this overlap will provide a clearer summary of modular transcriptional program we have uncovered.

Minor Concerns:

1) Can the authors also conduct RNAi of TfAP-2 and/or other TFs that they believe may specify T4/T5 sub-types to confirm some aspect of functional significance. This would be a substantive addition to the manuscript, elevating significance and impact, and they authors may likely have some of this data already in place?

A small number of TFs can define all eight individual T4/T5 subtypes: *bi, grn*, and *TfAP-2*. Here we have shown that *grn* completes a modular TF code that defines and directs axon targeting to all four LoP layers. We have also performed RNAi of the only remaining TF, *TfAP-2*, with multiple lines and did not observe significant knockdown of the protein (or targeting defects). Therefore, the role of *TfAP-2* in T4/T5 neurons currently remains inconclusive, and hence was omitted from our article. We have also found that removing *acj6*, a transcription factor common to all eight T4/T5 neurons, leads to massive disruption in the wiring patterns, including most importantly a complete loss of the four layers of axon terminals in the lobula plate. As subtype specification is the message in this paper, we do not think it is relevant to include these data.

2) It is difficult to understand if the grn LOF and GOF phenotype, and the previously published bi phenotype, fits with a combinatorial model for T4/T5 sub-type cell fate specification. Could the authors discuss this issue?

In this study we were able to dissect the combinatorial TF code into modular regulatory programs, and show how separate TFs can independently control distinct features of neural identity. Through unsupervised analysis, we were able to identify and link separate TFs to specific features of T4/T5 morphology (Figure 7). We have now expanded on how our phenotypes reflect modular control of wiring features in the Discussion.

Our results demonstrate a highly specific role of *grn* in axon targeting, supporting our hypothesis that separable transcriptional modules control discrete wiring features in T4/T5 subtypes, the primary conclusion of our paper. Here *grn* controls the development of “b/c”-subtype targeting, independently from axon targeting to the broad a/b and c/d LoP domains, and from dendrite targeting to M10 and Lo1. Both loss and gain of function *grn* reduce the number of targeting layer specificities from four to two in a predictable way. Together with earlier work on *bi*, which controls targeting to a/b versus c/d LoP domains (Apitz and Salecker, 2018), we have thus defined a complete modular TF code for T4/T5 axon targeting.

4) Figure 6: They use UAS-p35 to suppress the apoptosis triggered by grn overexpression, and could observe a similar phenotype even with apoptosis suppressed. Can they use the same approach to suppress apoptosis in grn RNAi, to see if this phenotype is also independent of apoptosis?

We agree this would be an excellent addition to our experiments. These experiments rely on multiple transgenes. While the location of the transgenes was compatible with the rescue of the death seen in the gain of function phenotype, it is not compatible with the loss of function. In short, three transgenes required for this experiment are inserted into the same chromosomal site in an autosome. While this is compatible for experiments requiring two of these (i.e. as it is for the rescue of the gain of function), it is not for the rescue of the RNAi loss of function.

5) When describing the T4/T5 neurons maybe they could elaborate more regarding the numbers of these neurons, and previous predictions of sub-types, based upon gene expression and morphology.

We have elaborated on numbers of T4/T5 neurons and references to previous sequencing efforts. In these studies, separate transcriptional programs for the eight individual morphological subtypes were not identified.

6) How does the 24h APF and 48h APF time-point used for scRNA-seq relate to when the T4/T5 neurons are born?

Neurogenesis of T4/T5 neurons occur early in late larval and early pupal stages of development (Pinto-Teixeira et al. 2018). By 24h APF all T4 and T5 neurons have already acquired their general morphologies, with incipient dendrites in medulla layer M10 and Lobula layer 1, respectively, and T4/T5 axon terminals in the LoP neuropil. At this stage, two layers are visible in the lobula. By 48h APF axon terminals sublaminate into four layers in LoP (Figure6-supplement 1) and the dendritic arbors are more complex. It is important to note that early stages of T4/T5 neurogenesis occurs as a wave giving rise to a gradient of neuronal maturation, similar to the generation of photoreceptors. At later stages, at least by 48h APF, development of T4 and T5 neurons appears to become synchronized in development.